# Vancomycin Drug Reaction with Eosinophilia and Systemic Symptoms: Meta-Analysis and Pharmacovigilance Study

**DOI:** 10.3390/jcm14030930

**Published:** 2025-01-31

**Authors:** Mohammed Aboukaoud, Yotam Adi, Mahmoud Abu-Shakra, Yocheved Morhi, Riad Agbaria

**Affiliations:** 1Department of Clinical Pharmacology, Faculty of Health Sciences, Ben-Gurion University, Beer Sheva 8410101, Israel; mahmouds2@clalit.org.il (M.A.-S.); riad@bgu.ac.il (R.A.); 2Faculty of Medical and Health Sciences, Tel-Aviv University, Tel-Aviv 6997801, Israel; yotamadi@mail.tau.ac.il; 3Soroka University Medical Center, Department of Internal Medicine D, Beer Sheva 8410101, Israel; 4Technion, The Ruth and Bruce Rapport Faculty of Medicine, Haifa 3525433, Israel; yocheved.m@campus.technion.ac.il

**Keywords:** vancomycin, antimicrobials, drug reaction with eosinophilia and systemic symptoms, HLA allele frequency, ethnicity

## Abstract

**Background**: Drug reaction with eosinophilia and systemic symptoms is a severe cutaneous reaction with a high mortality rate. It is challenging to diagnose due to its similar presentation to infectious disease syndromes, variation with the culprit drug, and lack of awareness. **Methods**: We searched PubMed, and Embase, for RegiSCAR-scored observational studies, the FDA Adverse Events Reporting System (FAERS) for adverse event reports, and the Allele Frequency Net Database (AFND) for HLA allele frequency. In our meta-analysis, we employed a random effects model to subgroup patients by ethnicity to determine the proportion of DRESS cases compared with various associated medications. Additionally, we identified a correlation between the proportion of cases and the presence of the HLA*A-32:01allele, which is suspected to predispose individuals to DRESS. **Results**: Twenty-one studies on 1949 DRESS cases in vancomycin and 2558 antimicrobial DRESS reports in the FAERS database were analyzed. Meta-analysis showed a 27% incidence of vancomycin-DRESS, with Caucasians having the highest proportion at 36%. The median latency for symptom onset was 21 days, with no female predisposition. The proportional incidence of vancomycin-DRESS did not correlate with the HLA-A*32:01 allele. The adjusted ROR for vancomycin was 2.40 compared to other antimicrobials, and the risk increased by 77% with concurrent antimicrobials. Piperacillin/tazobactam had a similar DRESS reporting risk at 0.95 (95%CI: 0.88–1.02). **Conclusions**: Vancomycin significantly contributes to the incidence of DRESS and is more closely related to ethnicity than to allele frequency, indicating that the HLA-A*32:01 allele may not be directly involved. Furthermore, the use of other antimicrobials can influence the reaction, underscoring the need to minimize antimicrobial use for better coverage.

## 1. Introduction

Drug reaction with eosinophilia and systemic symptoms (DRESS) is a severe cutaneous reaction that necessitates immediate attention and intervention. It is characterized by high fever, maculopapular rash, hematologic abnormalities (eosinophilia or atypical lymphocytes), and internal organ involvement. The incidence of DRESS is 1/1000–1/10,000, and the overall mortality rate is estimated at 2–10% [1]. It is suggested that patients may have a genetic predisposition due to variations in human leukocyte antigens (HLAs), which expose vancomycin with antigen-presenting cells and result in the activation of macrophages and T cells, coinciding with a type IV allergic reaction [2]. This involves CD4+ and CD8+ T cells, plasma dendritic cells, and monocytes enriched in the dermis, releasing TNF-α, IFN-γ, and interleukin 5 (IL-5). IL-5 is essential inducing the activation and migration of eosinophils that drive inflammation in DRESS [3]. In a small randomized controlled trial, the IL-5 axis blockade as a steroid-sparing agent demonstrated potential therapy for steroid-resistant cases [4].

DRESS occurrence might be associated with the human herpesvirus (HHV)-6 and 7, Epstein–Barr virus (EBV), and cytomegalovirus (CMV) reactivation [5].

Early recognition and intervention are crucial for ensuring the best possible outcome for the patient. One key challenge in diagnosing this syndrome is the lack of awareness among primary healthcare providers. This underscores the urgent need for healthcare providers to be more cognizant of DRESS. It is important to note that eosinophilia is not a reliable indicator as it may be absent in half of the patients. Thus, clinical manifestations are often mistaken for other infectious differential diagnoses [2].

Further, the affected organs and the long latency period vary with the culprit drug and range between three weeks and two months, complicating the causality assessment [6]. Early recognition, the discontinuation of the suspected drug, and the implementation of necessary supportive and specific measures are crucial. Systemic immunosuppression with glucocorticoids is usually first-line therapy; evidence suggests that cyclosporin may be a better alternative [7,8].

Vancomycin is indicated in treating sepsis and may overlap with the systemic manifestations of DRESS [9]. It is imperative to clearly outline the syndrome connected to vancomycin and thoroughly investigate the related risk factors in specific drugs, as DRESS characteristics change with the culprit drug. Our study combines various databases to investigate DRESS with vancomycin, including meta-analysis, the Allele Frequency Net Database (AFND), and the FDA Adverse Event Reporting System (FAERS). We aim to describe the syndrome’s characteristics in vancomycin and decipher the risk factors, strength, and correlation with the associated human leukocyte antigen (HLA) allele, DRESS frequency in different ethnicities, and the impact of concomitant use of antimicrobials.

## 2. Methods

The Allele Frequency Net Database (AFND) provides the scientific community with a repository for storing immune gene frequencies in different worldwide populations. Allele frequency is the number of allele copies in the population sample (Allele/2n) [10]. We queried the database for associated alleles with vancomycin and found that HLA-A*32:01 was the primary allele. When possible, we collected the HLA-A*32:01 allele frequency associated with vancomycin-DRESS from the included studies. For studies that did not determine the cohort’s allele frequency, we complemented the data based on ethnicity and study region using the AFND.

### 2.1. Meta-Analysis of Proportions

#### 2.1.1. Data Sources and Searches

We identified cohort studies investigating DRESS events in vancomycin by searching the PubMed and Embase databases. All eligible studies in English published up to August 2024 were considered. The literature searches used specific keywords related to vancomycin and DRESS. The complete search strategies, which incorporated index terms, Medical Subject Headings (MeSH), and text words for the search in question in the form of Population Intervention Comparison Outcome (PICO), are summarized in Appendix A. Given the rarity of DRESS, we included diverse sources in our study, such as cohort studies, surveys, conference abstracts, editorials, communications, and letters to the editor. This approach ensured that our research was inclusive. Case reports and pharmacovigilance studies were the only sources that we excluded. This meta-analysis followed the Preferred Reporting Items for Systematic Reviews and Meta-Analyses (PRISMA) guidelines, MOOSE (meta-analysis of observational studies in epidemiology) guidelines, and the ROBIN-I (risk of bias in non-randomized interventions) was used as a guideline [11,12]. The planned analysis was registered at the PROSPERO International Prospective Register of Systematic Reviews (CRD42024589196).

#### 2.1.2. Study Selection

Inclusion Criteria: Clearly described study region. DRESS diagnosis is based on RegiSCAR score criteria, including only probable and definite cases [13]. Sufficient description of the total medications used and known to be associated with DRESS, such as antiseizure medications and other antimicrobials. Demographics including ethnicity (Caucasians/Asian/Hispanic/Arab/Black). If specific details of ethnicity in vancomycin cases were reported as aggregates, we assumed based on the overall reported ethnicity in the study with a cutoff of >70%. The country where the events occurred was specified.

Exclusion Criteria: We excluded articles unavailable in English, case reports, pharmacovigilance analysis, and no reported drug triggers or their frequency. RegiSCAR with no cases (<2) or possible cases (2–3) were also excluded [13].

#### 2.1.3. Data Extraction and Quality Assessment

Three reviewers (M.A., Y.A., Y.M.) independently examined the articles for eligibility. We resolved disagreements by involving (R.A. and E.B-Y.). We assessed the risk of bias using the Joanna Briggs Institute (JBI) critical appraisal checklist for case series and calculated the score sum for each study from one to ten [14]. We assessed the certainty of evidence for the analysis using the Grading of Recommendations Assessment, Development, and Evaluation (GRADE) approach [15].

The reviewed observational studies determined the proportional incidence of vancomycin linked to DRESS among all drug-induced reported DRESS cases. We applied the meta-analysis model of effect sizes for estimating a single proportion and used the Clopper–Pearson method to compute the 95% confidence intervals (CI) provided in Stata version 18. We used the random effects model with the Der Simonian and Laird method to combine proportion estimates, considering the geographical differences and study characteristics. The studies were presented in forest plots. Heterogeneity was assessed using the I^2^ statistic, τ^2^, and Q test, with a significance level of *p* < 0.05. We performed a subgroup analysis to explore the proportion of vancomycin-associated DRESS in different ethnicities and studies, including >50% of females. We then narrowed the subgroup analysis to an HLA-A*32:01 allele frequency of >0.036, closer to the higher bound of HLA*A-32:01 frequency found in Caucasians, to test its effect. To study heterogeneity, we provided a meta-regression analysis of the studies’ ethnicity, allele frequency, sample size, age, and female proportion. Publication bias was assessed using a Doi plot; since funnel plots are inappropriate for assessing publication bias in a meta-analysis of proportions, Doi plots and the Luis Furuya-Kanamori (LFK) index are valuable alternatives [16,17].

Correlation Between HLA-A*32:01 and Ethnicity to the Proportion of Vancomycin-DRESS:

Beta regression was used to analyze the correlation between HLA-A*32:01 allele frequency and the proportion of vancomycin DRESS cases. All analyses were weighted by study size. This additional analysis aimed to support the above meta-analysis’s restricted subgroup analysis of >0.036.

### 2.2. Pharmacovigilance Analysis

A retrospective pharmacovigilance analysis was conducted using the FAERS database [18]. The analysis included patients reported as the primary suspect for DRESS events in their Medical Dictionary for Regulatory Activities (MedDRA) preferred terms between 2003 and 2024. The database was screened for DRESS reports in different antimicrobial groups associated with DRESS, including the following in their brand or generic names: vancomycin, sulfonamides, quinolones, ceftriaxone, ceftazidime, cefepime, and piperacillin/tazobactam. These antimicrobials were chosen because guideline-directed therapy recommends adding them to vancomycin in empirical coverage [9]. The complete search strategy is included in Appendix A.

For the FAERS analysis, we utilized a population-linkage program to identify and eliminate duplicate reports of the same drug–event combination. This was achieved by detecting matching information across seven key fields. Additionally, we collected data on age, sex, and the number of concomitant antimicrobials used for each report.

The FAERS database was used to analyze vancomycin’s disproportionality compared to other antimicrobials. Our logistic regression model adjusted for age, sex, and reports of additional antimicrobial use to detect signals for concomitant antimicrobial use in DRESS.

All statistical analyses with 95% CIs were performed using Stata version 18.

## 3. Results

### 3.1. Characteristics of Observational Studies

The literature search identified 953 studies; 932 did not meet the inclusion criteria. Figure 1 illustrates the PRISMA flow chart. Twenty-one studies describing 1949 DRESS cases were included; five were conducted in the US, four in Australia, three in Europe, two in Korea, one in Canada, one in Taiwan, one in India, one in Turkey, one in Tunisia, and one in Saudi Arabia. One included both European and US patients [19,20,21,22,23,24,25,26,27,28,29,30,31,32,33,34,35,36,37,38,39,40].

The total number of DRESS events was 1949, of which 484 were vancomycin-related. The mean age in vancomycin-DRESS was 47.5 ± 15, and a median of 40% (IQR: 34.5% to 55.5%) of females were included in the study. The median latency time to event was 21 days (IQR: 16.7–23) from the initiation of vancomycin. A total of 52% of the studies reported Drug-Induced Liver Injury (DILI) with DRESS (*p* = 0.01). The allele frequency ranged from 0.006 to 0.04. The study characteristics have been summarized in Table 1.

### 3.2. Proportion of Vancomycin-Induced DRESS

The overall proportion of vancomycin-related DRESS was 0.27 (95% CI; 0.20–0.36 I^2^ = 91%, *p* < 0.001) (Figure 2). The reported incidence of DRESS is 1/10,000 to 1/1000. Thus, the estimated incidence of vancomycin-DRESS would be 0.027 to 0.27 per 1000. The text refers to the higher bound of vancomycin-DRESS incidence, 0.27 per 1000.

In subgroup analysis, the proportion of vancomycin cases in studies including more than 50% females was lower, 0.18 (95% CI: 0.08–0.36, *p* < 0.001). However, the test for subgroup difference was not significant, at *p* = 0.18.

Restricting the subgroup analysis to an allele frequency greater than 0.036 demonstrated an overall proportion of 0.30 (95% CI 0.10–0.53, *p* < 0.001), with an insignificant subgroup difference from the pooled proportion (*p* = 0.99)—Appendix A.

Naturally, the heterogeneity was substantial. Meta-regression analysis identified ethnicity (R^2^ = 20%, Q = 12, df = 1, *p* = 0.001) and allele frequency (R^2^ = 27%, Q = 35, df = 1, *p* < 0.001) as sources of heterogeneity. Female proportion (*p* = 0.004) and study size (*p* = 0.01) also contributed, while age did not contribute to the heterogeneity of the studies.

#### 3.2.1. Quality Assessment

The JBI checklist for the case series assessed bias, resulting in a low risk with a median JBI score of 8 out of 10 (IQR 5.5–9.00) [14]. Appendix A shows a detailed risk assessment. This study’s certainty GRADE is very low, meaning that the authors’ confidence in the effect estimate is very low (Appendix A).

#### 3.2.2. Sensitivity Analysis

Sensitivity analysis demonstrated that the combined effect values remained consistent before and after excluding any study for the above outcomes. This consistency suggests that the study results were stable.

#### 3.2.3. Publication Bias

The Doi plot shows a symmetric distribution of studies, with a valid LFK index of 0.77 (values between +1 and −1 are considered symmetrical) [17]. The Eggers test demonstrated a *p*-value of 0.28, ruling out publication bias (Appendix A).

### 3.3. Beta Regression Analysis

We used beta regression to analyze the correlation between vancomycin-related DRESS cases and allele frequency. HLA-A*32:01 did not exhibit a significant correlation with the proportion of cases, *p*-value = 0.10, consistent with the subgroup analysis restricted to an allele frequency of >0.036. However, the proportion of DRESS cases and HLA frequency correlated positively and significantly with ethnicity at 0.58 (95%CI 0.15–1; *p*-value < 0.05) and 0.48 (95%CI; 0.23–0.73, *p*-value < 0.001), respectively.

### 3.4. FAERS Disproportionality Analysis

The FAERS database includes 29,153,222 reports. Between 2003 and 2024, 45,575 adverse events were reported in vancomycin, 2558 of which involved DRESS cases. The contrast arm of other antibiotics included 3833 DRESS reports, and 44% of the reports were associated with vancomycin compared to different antibiotics. The mean age of vancomycin cases was 50.9 ± 0.44, and 43.6% were females.

The median of the concomitant antimicrobials in the vancomycin group was significantly higher than that of the other antimicrobials.

The data detected a strong signal for vancomycin-related DRESS compared to all other drugs in the database (ROR: 80.3, 95% CI: 77.0–83.7).

Compared to the other antimicrobials tested, vancomycin had a significantly higher reporting risk (adj.ROR: 2.40, 95% CI; 2.27–2.55) but did not differ from piperacillin/tazobactam (adj.ROR: 0.95, 95% CI: 0.88–1.02).

The presence of concomitant antimicrobials significantly increased the reporting risk by 77% (adj.ROR: 1.77, 95% CI; 1.72–1.83). For most antimicrobials tested, the reporting risk was lower in females, and age did not influence the model. Table 2 shows the crude and adjusted analysis.

## 4. Discussion

This study is the first to use a combination of databases and various analytical tools, such as meta-analysis, pharmacogenetics, and pharmacovigilance, to detail and quantify the proportion of vancomycin-associated DRESS and the risk factors associated with this adverse event.

In our meta-analysis, 27% of DRESS cases were related to vancomycin compared to other associated medications, and the FAERS analysis found 44% of vancomycin-associated instances compared to different antimicrobials. The median latency period was 21 days from the index date of vancomycin.

The meta-analysis, supported by the FAERS analysis, did not find a female predisposition to the event. Age did not change the effect in either analysis, and there is a positive correlation between vancomycin DRESS events and Drug-Induced Liver Injury (DILI).

Caucasians carried the highest proportion of DRESS; Australians demonstrated the highest contribution at 40%. The Arab proportion was 16%, with Tunisians contributing the most at 30% compared to Saudis with Bedouin ancestors. This sparked a debate about possible Caucasian ancestry among Tunisians [39,40].

The correlation between vancomycin-associated DRESS and ethnicity was statistically significant. However, there was no apparent correlation between HLA-A*32:01 frequency and the proportion of vancomycin-induced DRESS, both in the restricted subgroup and beta-regression analyses, casting doubt on its relationship to DRESS.

Our findings suggest that HLA-A*32:01 may be incidental since the allele frequency correlated with ethnicity but not the proportion of cases. However, the under-reporting of DRESS and the frequency of population HLA typing still limits this conclusion.

It could be inferred from these findings that ethnicity or geographic-related factors other than HLA-A*32:01 influence a proportion of DRESS cases, such as infectious disease epidemiology, antimicrobial consumption, drug interactions, other unrecognized HLA alleles, or immune system mutations that trigger viral reactivation in disease states, specifically HHV-6, which is strongly associated with DRESS.

Since DRESS is often linked to the reactivation of latent viruses, such as HHV-6, Epstein–Barr virus (EBV), and cytomegalovirus (CMV), there are also documented ethnic differences in the prevalence of various viral infections. For instance, non-Western ethnic groups tend to have higher seroprevalence rates of cytomegalovirus (CMV), Epstein–Barr virus (EBV), and herpes simplex virus type 1 (HSV-1) among children [41]. These differences can be partly attributed to socioeconomic and environmental factors. Additionally, genetic ancestry can influence the immune response to viral infections, such as variations in the type I interferon pathway activity observed between individuals of European and African ancestry, which may affect outcomes in viral infections shown in influenza and COVID-19 [42]. Other factors that affect the immune response could be differences in lifestyle and diet between ethnic groups.

The evidence required to meet the Clinical Pharmacogenetics Implementation Consortium (CPIC) recommendations to conduct an HLA-A*32:01 screening before starting vancomycin therapy to prevent DRESS in certain ethnicities is substantially high [43]. For instance, the CPIC recommends screening for HLA-B*57:01 in abacavir hypersensitivity reaction was notably significant in the screening group (38 per 1000) compared to the control group (78 per 1000) [44]. Likewise, screening for HLA-B*15:02 in Taiwanese patients reduced the incidence of carbamazepine-induced Steven Johnson’s syndrome to 3.3 cases per 1000 from 25 per 1000 [45].

These reported incidences for abacavir and carbamazepine are higher than the estimated upper bound incidence of vancomycin-induced DRESS (0.27 per 1000), probably because of greater penetrance and significant correlation with the associated HLA allele, which illustrates the issue’s complexity [43,46,47]. The benefits of HLA testing in abacavir and carbamazepine are more apparent than those associated with vancomycin and HLA-A*32:01.

Two small studies conducted in China suggest a higher risk of vancomycin-DRESS in HLA-A*32:01 carriers. However, since the cohorts in these studies are limited and solely derived from China, we cannot exclude other ethnic factors that may influence these findings [26,34].

Disproportionality analysis reveals that the adjusted reporting risk for vancomycin is twice as high as for other antimicrobials and is highest compared to quinolones, suggesting a lower ratio in quinolones. Conversely, the lowest reporting risk is for sulfonamides, indicating a lower ratio of DRESS with sulfamethoxazole than the other antimicrobials, as per Sharifzadeh et al.’s review [5]. Vancomycin did not differ from piperacillin/tazobactam, suggesting that both have a similar risk of DRESS.

Notably, vancomycin had a higher median of concomitant antimicrobials since it covers only Gram-positive bacteria. This prompts clinicians to use additional antimicrobials for Gram-negative bacteria in empirical therapy. Our model accounted for the number of concomitant antimicrobials.

This study demonstrated that adding antimicrobials to vancomycin significantly increased the reporting risk of DRESS by 77%. This is intriguing because sulfonamides, cephalosporins, and quinolones do not interfere with the pharmacokinetics of vancomycin; the mechanism may be purely pharmacodynamic [48]. Increased concomitant antimicrobials may influence DRESS in vancomycin rather than just HLA*A-32:01 frequency, thus minimal antimicrobial use is prudent to obtain maximum coverage.

The primary clinical implications of our study are to raise awareness regarding drug reaction with eosinophilia and systemic symptoms (DRESS) associated with vancomycin. Our findings suggest that vancomycin may be strongly associated with DRESS, whereas the frequency of HLA-A*32:01 may not have a direct role in this context and could be considered a bystander. Conversely, concurrent use of antimicrobial agents emerged as a significant risk factor. Based on our results, we did not find that HLA-A*31:01 is a suitable candidate for pharmacogenetic implications. Integrating this differential diagnosis as early as possible remains crucial for facilitating the appropriate treatment and mitigating the risk of escalation.

### Strengths and Limitations

This is the first study to utilize various databases to gain insights into vancomycin-associated DRESS. It comprises 1949 cases from observational studies and 6391 from the FAERS database. We included data from different countries. Even though limited information was available on the issue, this study used strict criteria for including and excluding data. The JBI score for the risk of bias indicated a low risk. Publication bias was not detected.

Our study has limitations. Plasma levels of vancomycin and the duration of vancomycin treatment were not reported, limiting our analysis for these variables. Only one study included black ethnicity. Although the overall heterogeneity of the studies is high, 47% can be explained by expected variables such as ethnicity and allele frequency and could be attributed to incorporating diverse data sources, considering the rarity of DRESS. Sample sizes can be a significant source of heterogeneity and may introduce bias, as small samples can lead to substantial changes in estimates with minor variations in cases. Additionally, smaller studies are more susceptible to selection or recall bias. However, regarding sample size, we conducted a FAERS analysis that provided usage with larger data.

The last analysis of the FAERS database has several limitations, particularly in establishing the timing and causality when adjusting for concurrent antimicrobial use. However, with the extensive number of cases and long-term follow-up, analyzing the post-marketing adverse events database is crucial for identifying such rare adverse events.

## 5. Conclusions

This study is the first to utilize meta-analysis, pharmacovigilance, and pharmacogenetic databases to investigate the proportion of vancomycin-induced DRESS and its correlation with ethnicity and HLA-A*32:01 allele frequency. This study’s findings reveal that vancomycin is linked to 27% of DRESS cases, with Caucasians showing the highest proportion among the ethnicities studied. Moreover, the frequency of the HLA-A*32:01 allele did not correlate significantly with the proportion of DRESS cases, despite ethnicity correlating with the proportion of DRESS cases. These findings suggest that factors related to ethnicity, apart from HLA-A*32:01, may influence the proportion of DRESS. Additionally, using concomitant antimicrobials emerges as a substantial factor in DRESS. These results underscore the importance of exercising prudence when using minimal combinations of antimicrobials that provide maximal coverage.

## Figures and Tables

**Figure 1 jcm-14-00930-f001:**
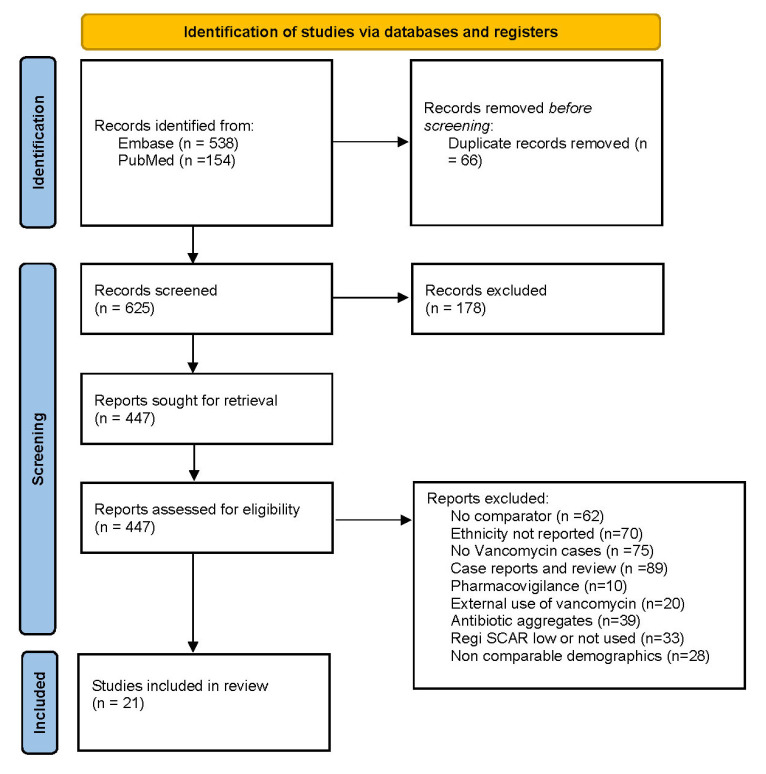
Search strategy illustrated by the PRISMA flow chart.

**Figure 2 jcm-14-00930-f002:**
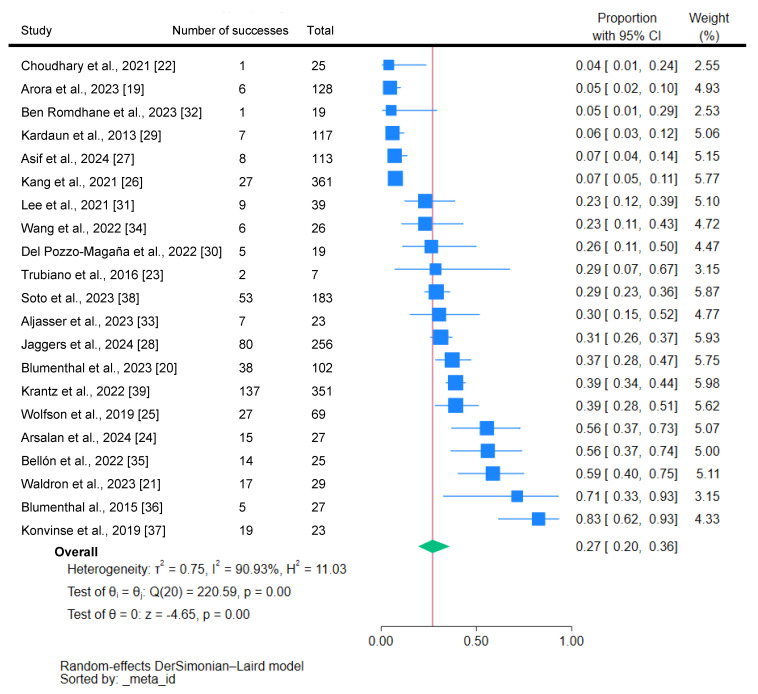
Proportion of vancomycin associated with drug reaction with eosinophilia and systemic symptoms. In subgroup analysis, Caucasians had the highest proportion of vancomycin-DRESS at 0.36 (95% CI: 0.27–0.45); among them, Australians had the highest proportion at 0.40 followed by Canadians at 0.26, Americans at 0.23, and Europeans at 0.21. Arabs at 0.16 (95% CI: 0.03–0.53). Asians and Black ethnic groups had the lowest proportions at 0.13 (95% CI: 0.06–0.27) and 0.05 (95% CI: 0.02–0.10), respectively (Figure 3) [19,20,21,22,23,24,25,26,27,28,29,30,31,32,33,34,35,36,37,38,39].

**Figure 3 jcm-14-00930-f003:**
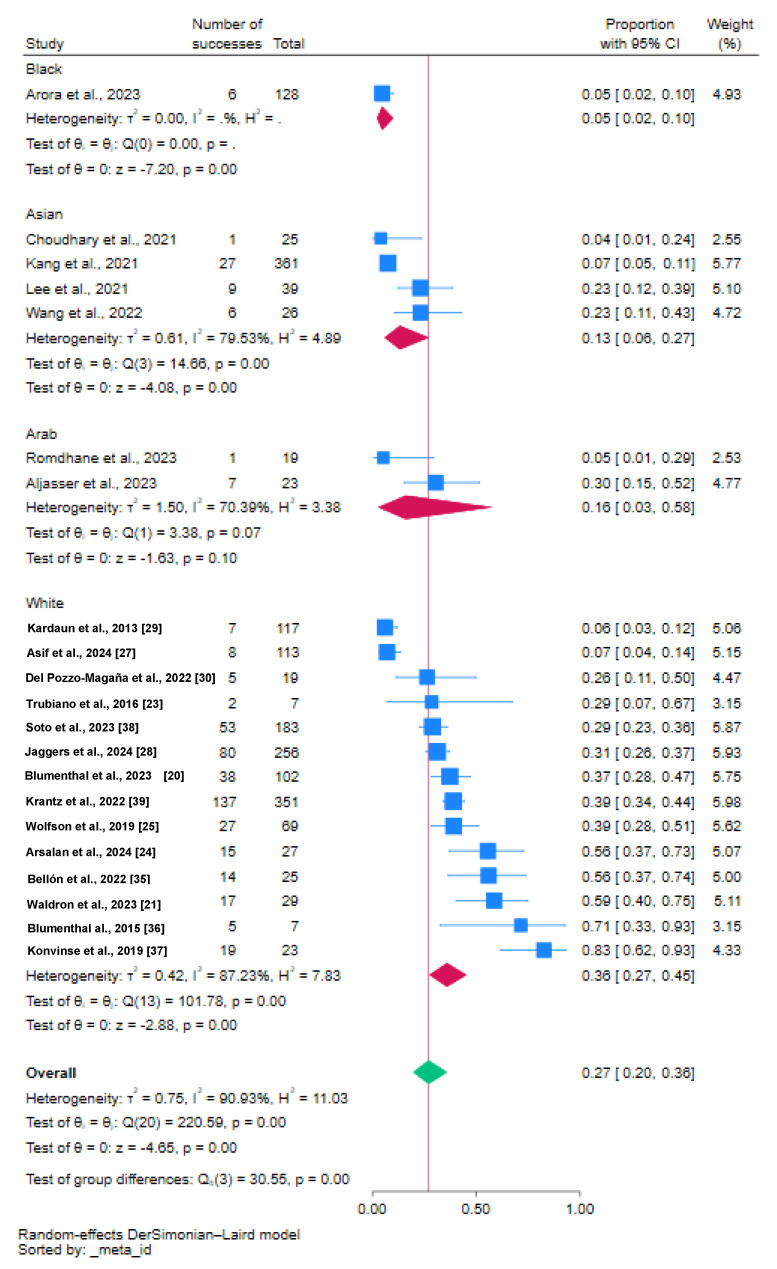
Proportion of vancomycin-associated DRESS by ethnicity [19,20,21,22,23,24,25,26,27,28,29,30,31,32,33,34,35,36,37,38,39].

**Table 1 jcm-14-00930-t001:** Selected meta-analysis study characteristics.

Study	Overview	Country	Age (Y)	Female Sex (%)	Ethnicity	Haplotype and Frequency	Sample Size	Limitations
Arora et al., 2023 [19]	Drug-Induced Liver Injury Network [DILIN], events defined as DRESS with DILI.	US	42.3	59	Black	NR (HLA-A*32:01, 3.8%, 0.02)	128	No concomitant antibiotics or comorbidities were reported.
Blumenthal et al., 2023 [20]	Research letter electronic health records.	Europe	52	63	Caucasians	NR (HLA-A*32:01, 0.019–0.072)	102	No concomitant antibiotics or comorbidities were reported. Specific populations in Europe were not stated.
Waldron et al., 2023 [21]	Abstract two prospective multicenter cohorts.	Australia	NR	NR	Caucasians	HLA-A*32:01 (0.0340)	29	Lack of demographic data.
Choudhary et al., 2021 [22]	Observational prospective tertiary care center.	India	44.3	NR	Asian	NR (HLA-A*32:01, 7.8%, 0.0390)	25	Lack of demographic data.
Trubiano et al., 2016 [23]	Retrospective cohort study inpatient study.	Australia	57	40	Caucasians	NR (HLA-A*32:01, 0.0340)	84	Median no. of concomitant antibiotics was 3.
Arsalan et al., 2024 [24]	A retrospective study was conducted at a tertiary-level university hospital.	Turkey	10	34	Caucasians	NR (HLA-A*32:01, 0.0490)	27	Vancomycin and meropenem or Ceftazidime.
Wolfson et al., 2019 [25]	Registry of severe cutaneous adverse reactions. Partners Healthcare System (PHS), Boston.	Europe and US	60	55	Caucasians	NR (HLA-A*32:01, 0.036)	69	55% included Vancomycin alone.
Kang et al., 2021 [26]	A nationwide registry by the Korean SCAR consortium.	Korea	63	50	Asian	NR (HLA-A*32:01, 0.0061)	361	HLA-A*32:01, the genetic risk marker of vancomycin-induced DRESS, is rare (0.3%) in Koreans, but other genetic risk factors were not stated.
Asif et al., 2024 [27]	Observational, DILIN National Institutes of Health.	US	60	33	Caucasians	HLA-A*32:01, 78% of cases, (0.022)	113	3 cases were judged to be highly likely due to vancomycin-carried HLA-A*32:01 and 4 of the 6 cases (67%) were judged to be probably due to vancomycin. Allele frequency was inferred by the author’s location.
Jaggers et al., 2024 [28]	Clinical communication. Health-related quality of life questionnaire	US	46	40	Caucasians	NR (HLA-A*32:01, 4.4%, 0.022)	256	29% were non-responders. Concomitant medications are not stated.
Kardaun et al., 2013 [29]	Prospective multinational registry of severe cutaneous adverse reactions (SCAR).	Austria, England, France, Germany, Israel, Italy, Taiwan, and The Netherlands	48	20	Caucasians	NR (HLA-A*32:01, 5.5–13.8%, 0.027–0.050)	117	The number of cases in Taiwanese was not stated, but the majority were European. The median concomitant medications were 4. The number of suspect drugs was substantially reduced after elimination due to the time course.
Del Pozzo-Magaña et al., 2022 [30]	Retrospective study of all cases of DRESS admitted to the Institution.	Canada	59	48	Caucasians	NR (HLA-A*32:01, 5.6%, 0.0278)	19	One patient used concomitant ceftazidime with vancomycin.
Lee et al., 2021 [31]	Retrospective study of medical records in patients with HLA results using SUPREME^®^, a clinical data warehouse of the Seoul National University Hospital (SNUH).	Korea	44	37	Asian	HLA-A*32:01, 1%, (0.0061)	11,998	DRESS inferred but not explicitly stated.
Ben Romdhane et al., 2023 [32]	Retrospective analysis of all cases of DRESS is diagnosed in pediatric patients (age ≤ 18 years).	Tunisia	7	100	Arab	NR (HLA-A*32:01, 3%, 0.026)	19	Concomitant with Ceftriaxone.
Aljasser et al., 2023 [33]	A cross-sectional study conducted at King Abdulaziz Medical City, Riyadh.	Saudi Arabia	41	47.8	Arab	NR (HLA-A*32:01, 4.7%. 0.024)	23	There was one case of SJS-TEN and DRESS-TEN overlap. Concomitant medications are not stated.
Wang et al., 2022 [34]	Retrospectively enrolled from the SCAR consortium.	Taiwan	57	19	Asian	HLA-A*32:01, 7.7%, (0.0040), HLA-B*40:06, 11.5%, (0.02), HLA-B*67:01, 7.7%, (0.0060), HLA-B*07:05, 3.8%, (0.001)	26	6 out of 26 subjects were concomitantly receiving other medicines (including amoxicillin, ceftriaxone, teicoplanin, valproic acid, diclofenac, and esomeprazole) when prescribed with vancomycin.
Bellón et al., 2022 [35]	Case–control study of Spanish registry PIELenRed.	Europe-Spain	44	36	Caucasians	HLA-A*32:01, 4%, (0.040)	25	Latency was inferred from non-events. Demographic characteristics were lacking, but 50% of DRESS vancomycin cases carried HLA-A*32:01 compared with 4%, OR: 13.33 (1.36–130.30)
Blumenthal et al., 2015 [36]	Cohort study of inpatients Massachusetts General Hospital (Boston, Massachusetts).	US	64	48	Caucasians	NR (HLA-A*32:01, 0.036)	210	Ethnicity was inferred from the cohort used. The proportion of ethnic origins was not stated.
Konvinse et al., 2019 [37]	Retrospective study of patients detected by using Vanderbilt’s BioVU repository, a deidentified electronic health record (EHR) database linked to a DNA biobank.	Australia	51	38	Caucasians	HLA-A*32:01, 6.8%, (0.034)	54,249	The study reported cases of vancomycin DRESS in HLA-A*32:01 carriers vs. no carriers. Two patients were on vancomycin alone.
Soto et al., 2023 [38]	Abstract of a retrospective study at Mass Bringham.	US	50	58	Caucasians	NR (HLA-A*32:01, 0.036)	183	66% were Caucasians; the other 34% ethnicity was not stated.
Krantz et al., 2022 [39]	Retrospective DRESS cases from the Synthetic Derivative (SD) of Vanderbilt University Medical Center (VUMC).	Australia	48.5	56	Caucasians	NR (HLA-A*32:01, 0.026)	351	Ethnicity was not reported but was inferred from the location of the hospital.

NR = not reported, HLA = human leukocyte antigen.

**Table 2 jcm-14-00930-t002:** Crude and adjusted reporting odds ratio for vancomycin DRESS events compared to other antimicrobials.

Comparator	Crude Reporting Odds Ratio (95%CI)	Adjusted Reporting Odds Ratio (95%CI)	Effect of Concomitant Antibiotics (95%CI)	Female Sex (95%CI)	*p*-Value
Composite Other Antimicrobials	2.52 (2.39–2.66)	2.40 (2.27–2.55)	1.77 (1.72–1.83)	0.83 (0.79–0.88)	<0.0001
Sulfonamides	1.98 (1.85–2.10)	1.68 (1.57–1.80)	1.66 (1.59–1.73)	0.74 (0.69–0.77)	<0.0001
Third Generation Cephalosporins	1.79 (1.64–1.94)	2.99 (2.72–3.28)	1.81 (1.74–1.89)	0.97 (0.89–1.04)	<0.0001
Quinolones	4.47 (4.11–4.86)	3.67 (3.35–4.02)	1.73 (1.66–1.81)	0.91 (0.84–0.98)	<0.0001
Piperacillin/tazobactam	1.05 (0.98–1.13)	0.95 (0.88–1.02)	1.89 (1.81–1.97)	0.99 (0.92–1.06)	0.17

## Data Availability

The original data presented in the study are openly available in Pubmed references 20–40. Adverse event reports are found in the FAERS repository https://www.fda.gov/drugs/fdas-adverse-event-reporting-system-faers/fda-adverse-event-reporting-system-faers-public-dashboard, accessed on 13 January 2025. HLA allele frequency is found at the AFND http://www.allelefrequencies.net/, accessed on 13 January 2025.

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
