# Peer review of "Vancomycin Drug Reaction with Eosinophilia and Systemic Symptoms: Meta-Analysis and Pharmacovigilance Study"

_jcm, 2025, doi:10.3390/jcm14030930_

Round 1

Reviewer 1 Report

Comments and Suggestions for Authors

Comments to authors

The authors carried out a prevalence study of DRESS with the use of vancomycin and also analysed the possible influence of certain factors that could be considered risk factors. Although the manuscript is interesting, it needs some improvements. In particular, the structure of the methodology, results and discussion, in addition to expanding the discussion, specifically on the possible influence of the sample size of the meta-analysis on the observed heterogeneity. Finally, I suggest reviewing the use of the GRADE tool, it is surprising that with observational studies and high inconsistency it comes out as moderate.

Abstract

Ok

Introduction

·            Firstly, the introduction is one paragraph. Please split it into three paragraphs, it will be easier to read.

·            A mention of the mechanism by which antibiotics, or in this case vancomycin, can cause DRESS would be appreciated.

Methods

·            The different methods sections (search strategy, study selection, study quality assessment, etc.) should be included in their respective subsections.

·            If the meta-analyses are observational studies, it should be stated and cited that the MOOSE guide and the Cochrane Collaboration handbook (in addition to PRISMA) were followed.

Results

·            Table 1: The Overview column is not aesthetically attractive. It should be split into two columns: design and country.

·            Also, in Table 1, it would be nice to include the citation for each study in the Study column.

·            It is better to refer to them as Caucasians rather than White (throughout the manuscript).

·            “The overall proportion of vancomycin-related DRESS was 27%”: If we are talking about the proportion, it should be 0.27. In prevalence it would be 27%. I suggest putting it as a proportion and in the discussion as a prevalence.

·            GRADE: Check the GRADE rating. You get "moderate", but for me, with the same data (including the fact that they are observational studies), I get "very low".

·            "3.1.1 Beta regression analysis": shouldn't it be 3.3?

·            Subsections such as "quality assessment", "sensitivity analysis", etc. should be listed as separate subsections.

Discussion

·            The beginning of the discussion is very much in need of improvement (it then improves a little as it progresses). The first paragraph should be a summary of the findings. Then several paragraphs explaining and contextualising the findings. Then a paragraph on clinical implications and finally a paragraph on limitations.

·            There is one limitation that is not mentioned, but which should also be discussed in the discussion as a possible source of heterogeneity, and that is the sample size of some studies. Small sample sizes tend to overestimate the estimated outcome. For example, in Figure 2, the last 5 studies have sample sizes of less than 30 participants. There are two reasons for this possible bias in the estimates. First, with small samples, a small variation in cases changes the estimate significantly. Secondly, small studies are more likely to suffer from selection or recall bias.

References

·            Some citations may have errors in form (not content). Check citation 9, 12, etc.

·            The PROSPERO citation is not necessary, it is sufficient to include the PROSPERO code in the manuscript.

·            Reference 16 should be changed: there are peer-reviewed articles that can be cited.

Author Response

General Author's Response:

We sincerely appreciate the reviewer for their time and effort in reviewing our paper. Their insightful comments have been invaluable in helping us enhance the quality of our research significantly. Thank you for your constructive feedback!

Comment 1 Introduction: Firstly, the introduction is one paragraph. Please split it into three paragraphs; it will be easier to read.

Author's Reply: Thank you for the remark; the paragraph was split.

 Comment 2: A mention of the mechanism by which antibiotics, or in this case vancomycin, can cause DRESS would be appreciated.

Author's Reply: Indeed, this adds majorly to our introduction and was mistakenly elaborated on in the discussion.

Line 45-53, Page 2: the introduction of the mechanism was added as follows; “It is suggested that patients may have a genetic predisposition due to variations in human leukocyte antigens (HLA), which expose vancomycin by antigen-presenting cells and result in the activation of macrophages and T cells, coinciding with a type IV allergic reaction.”

A deeper understanding of the mechanisms is also proposed in the first paragraph of the discussion, which was replaced to the introduction. 

Comment 3 Methods:  The different methods sections (search strategy, study selection, study quality assessment, etc.) should be included in their respective subsections.

Author's Reply: The methods were reorganized in such a way that the relevant paragraphs are found to coincide with the methodology incorporated. Pages: 2-4.

Line 74: “The Allele Frequency Net Database (AFND)….”

Line 85: “Meta-analysis of Proportions:….”

Line 101: “Correlation Between HLA-A*32:01 and Ethnicity to the Proportion of Vancomycin-DRESS:…”

Line 139: “Pharmacovigilance Analysis:….”

Comment 4:   If the meta-analyses are observational studies, it should be stated and cited that the MOOSE guide and the Cochrane Collaboration handbook (in addition to PRISMA) were followed.

Author's Reply:  We agree, thank you for pointing this out. We added a consideration to the ROBIN-I guidelines used.

Line 96-100, Page 3: “This meta-analysis followed the Preferred Reporting Items for Systematic Reviews and Meta-Analyses (PRISMA) guidelines, and the ROBIN-I (risk of bias in non-randomized interventions) was used as a guideline [8]…..” The planned analysis was registered at the PROSPERO International Prospective Register of Systematic Reviews (CRD42024589196).”

Comment 5:  Table 1: The Overview column is not aesthetically attractive. It should be split into two columns: design and country.

Author's Reply: We agree! Table 1 was re-organized as suggested. Pages: 6-11

Comment 6:   It is better to refer to them as Caucasians rather than White (throughout the manuscript).

Author's Reply: Thank you for the remark. The terminology was changed adequately to coincide with your comment. 

Comment 7: 

  “The overall proportion of vancomycin-related DRESS was 27%”: If we are talking about the proportion, it should be 0.27. In prevalence it would be 27%. I suggest putting it as a proportion and in the discussion as a prevalence.

Author's Reply: Thank you for bringing this error to our attention. 27% was changed to 0.27 as requested. Line 195-198. 

Comment 8: GRADE: Check the GRADE rating. You get "moderate", but for me, with the same data (including the fact that they are observational studies), I get "very low".

Author's Reply: Thank you for pointing out this major misunderstanding. 

The certainty grade was re-evaluated to very low. Very Low: Very little confidence in the effect estimate, incorporated in the text line: 225-226. Page 13.

The correction was added to the supplementary table S3. 

Comment 8: "3.1.1 Beta regression analysis": shouldn't it be 3.3?

Author's Reply: Indeed! Corrected. Thank you! 

Comment 9:      The beginning of the discussion is very much in need of improvement (it then improves a little as it progresses). The first paragraph should be a summary of the findings. Then several paragraphs explaining and contextualising the findings. Then a paragraph on clinical implications and finally a paragraph on limitations.

Author's Reply: the first paragraph was moved to the introduction, so the discussion begins with a summary of our findings. 

Comment 10:  There is one limitation that is not mentioned, but which should also be discussed in the discussion as a possible source of heterogeneity, and that is the sample size of some studies. Small sample sizes tend to overestimate the estimated outcome. For example, in Figure 2, the last 5 studies have sample sizes of less than 30 participants. There are two reasons for this possible bias in the estimates. First, with small samples, a small variation in cases changes the estimate significantly. Secondly, small studies are more likely to suffer from selection or recall bias.

Authour's Reply: Thank you for this comment, we agree and have adopted your proposal. This is one of the reasons we chose to introduce an additional real-world analysis from the FDA adverse event reporting system.

Line 349-353 : "Sample sizes can be a significant source of heterogeneity and may introduce bias, as small samples can lead to substantial changes in estimates with minor variations in cases. Additionally, smaller studies are more susceptible to selection or recall bias. However, regarding sample size, we conducted a FAERS analysis that provided use with larger data."

Thank you for pointing out the errors in some citations, we revised our references and made corrections in some of the citations. 

Reviewer 2 Report

Comments and Suggestions for Authors

It is noteworthy that this study showed that ethnicity is more involved in vancomycin-associated DRESS than the association with HLA-A*32:01.

HLA proteins play a crucial role in the immune system by presenting foreign substances (like drugs) to T cells. Genetic variations in HLA genes can alter the way these proteins interact with drugs, potentially leading to an exaggerated immune response and triggering DRESS. HLA-A*32:01 proteins are expressed on the surface of most cells in the body. Their primary function is to present small fragments of proteins (called peptides) to CD8+ T cells, a type of white blood cell crucial for the immune response. However, if ethnicity rather than genetic polymorphism is the contributing factor, there may be ethnic differences in viral infection related to the mechanism of DRESS.DRESS is often associated with the reactivation of latent viruses, such as Epstein-Barr virus (EBV) or cytomegalovirus (CMV). The underlying immune dysregulation caused by the drug may allow these viruses to reactivate and contribute to the symptoms of DRESS. Consideration of whether there are any data on ethnic differences in the prevalence of viral antibodies would aid the authors' interpretation.

I think it's a good idea to focus on viral infection as a factor explaining ethnic differences in the incidence of DRESS, but it would also be good to add other factors that affect the immune response, such as differences in diet between ethnic groups.

Author Response

Reviewer Comments:

It is noteworthy that this study showed that ethnicity is more involved in vancomycin-associated DRESS than the association with HLA-A*32:01.

HLA proteins play a crucial role in the immune system by presenting foreign substances (like drugs) to T cells. Genetic variations in HLA genes can alter the way these proteins interact with drugs, potentially leading to an exaggerated immune response and triggering DRESS. HLA-A*32:01 proteins are expressed on the surface of most cells in the body. Their primary function is to present small fragments of proteins (called peptides) to CD8+ T cells, a type of white blood cell crucial for the immune response. However, if ethnicity rather than genetic polymorphism is the contributing factor, there may be ethnic differences in viral infection related to the mechanism of DRESS.DRESS is often associated with the reactivation of latent viruses, such as Epstein-Barr virus (EBV) or cytomegalovirus (CMV). The underlying immune dysregulation caused by the drug may allow these viruses to reactivate and contribute to the symptoms of DRESS. Consideration of whether there are any data on ethnic differences in the prevalence of viral antibodies would aid the authors' interpretation.

I think it's a good idea to focus on viral infection as a factor explaining ethnic differences in the incidence of DRESS, but it would also be good to add other factors that affect the immune response, such as differences in diet between ethnic groups

Author's Reply: Thank you for your time and effort in reviewing our manuscript. The explanation of the mechanism and suggestion of ethnicity related factors other then HLA-A*32:01 provided us with an open mind on this issue. We agree that the emphasis should be on those related factors and thus the following was added to the discussion lines: 303-312, Page 15. 

"DRESS is often linked to the reactivation of latent viruses, such as HHV-6, Epstein-Barr virus (EBV), and cytomegalovirus (CMV). There are also documented ethnic differences in the prevalence of various viral infections. For instance, non-Western ethnic groups tend to have higher seroprevalence rates of cytomegalovirus (CMV), Epstein-Barr virus (EBV), and herpes simplex virus type 1 (HSV-1) among children. These differences can be partly attributed to socioeconomic and environmental factors. Additionally, genetic ancestry can influence the immune response to viral infections. Variations in the activity of the type I interferon pathway have been observed between individuals of European and African ancestry, which may affect outcomes in infections such as influenza and COVID-19."  

"Other factors that affect the immune response, could be differences in life-style and diet between ethnic groups."

Round 2

Reviewer 1 Report

Comments and Suggestions for Authors

Comments to authors

The authors have correctly addressed most of the issues raised. However, I suggest three improvements, two in form and one in content, as detailed below.

Methods

·            Please include and cite the MOOSE (Meta-analysis of observational studies in epidemiology) guidelines and the Cochrane Collaboration Handbook (not just PRISMA) in your methods.

Results

·            Maybe I didn't explain myself well. Always report the estimates in the results as proportions, not as prevalences, i.e. include the estimates as they appear in the forest plots.

Discussion

·            “The main clinical implication is considering minimal antimicrobials to obtain maxi-332 mum coverage, thus advocating prudent antimicrobial use.”: It is clear that antibiotics should only be used when necessary. I am sure your study has further implications. Or at least deeper ones.

Author Response

Thank you for this valuable second round revision and we apologies for any inconvenience. 

Comment 1: Please include and cite the MOOSE (Meta-analysis of observational studies in epidemiology) guidelines and the Cochrane Collaboration Handbook (not just PRISMA) in your methods.

MOOSE guideline citation was added in line: 102-103. 

Comment 2: Maybe I didn't explain myself well. Always report the estimates in the results as proportions, not as prevalences, i.e. include the estimates as they appear in the forest plots.

We apologize for our misunderstanding. Percentages were converted to proportions pages 10-11.

Comment 3:    “The main clinical implication is considering minimal antimicrobials to obtain maxi-332 mum coverage, thus advocating prudent antimicrobial use.”: It is clear that antibiotics should only be used when necessary. I am sure your study has further implications. Or at least deeper ones.

Thank you for pointing out clarifying the importance of the take home messages in the text. The following clinical implications has been added. Line: 333-342, page 16.

"The primary clinical implications of our study are to raise awareness regarding Drug Reaction with Eosinophilia and Systemic Symptoms (DRESS) associated with vancomycin. Our findings suggest that vancomycin may be strongly associated with DRESS whereas, the frequency of HLA-A*32:01 may not have a direct role in this context and could be considered a bystander. Conversely, concurrent use of antimicrobial agents emerged as a significant risk factor. Based on our results, we did not find that HLA-A*31:01 is a suitable candidate for pharmacogenetic implications. Integrating this differential diagnosis as early as possible remains crucial for facilitating appropriate treatment and mitigating the risk of escalation."